# On-Policy Optimization with Group Equivalent Preference for Multi-Programming Language Understanding

**Haoyuan Wu**[1*]   **Rui Ming**[1*]   **Jilong Gao**[1*]   **Hangyu Zhao**[1]   **Xueyi Chen**[1]
**Yikai Yang**[1]   **Haisheng Zheng**[2]   **Zhuolun He**[1,2]   **Bei Yu**[1]
[1]The Chinese University of Hong Kong    [2]ChatEDA Tech
{hywu24, byu}@cse.cuhk.edu.hk

## Abstract

Large language models (LLMs) achieve remarkable performance in code generation tasks. However, a significant performance disparity persists between popular programming languages (e.g., Python, C++) and others. To address this capability gap, we leverage the code translation task to train LLMs, thereby facilitating the transfer of coding proficiency across diverse programming languages. Moreover, we introduce OORL for training, a novel reinforcement learning (RL) framework that integrates on-policy and off-policy strategies. Within OORL, on-policy RL is applied during code translation, guided by a rule-based reward signal derived from unit tests. Complementing this coarse-grained rule-based reward, we propose Group Equivalent Preference Optimization (GEPO), a novel preference optimization method. Specifically, GEPO trains the LLM using intermediate representations (IRs) groups. LLMs can be guided to discern IRs equivalent to the source code from inequivalent ones, while also utilizing signals about the mutual equivalence between IRs within the group. This process allows LLMs to capture nuanced aspects of code functionality. By employing OORL for training with code translation tasks, LLMs improve their recognition of code functionality and their understanding of the relationships between code implemented in different languages. Extensive experiments demonstrate that our OORL for LLMs training with code translation tasks achieves significant performance improvements on code benchmarks across multiple programming languages.

## 1 Introduction

With the rapid advancement of LLMs [2], code-specific language models have garnered significant attention within the research community. Building upon pre-trained LLMs, code LLMs such as the StarCoder series [12, 16], CodeLlama series [23], DeepSeekCoder series [27], QwenCoder [20] series, and CodeStral [18] have demonstrated superior performance in code generation tasks. However, although current state-of-the-art code generation LLMs excel at generating code using popular programming languages (e.g., Python, C++), their performance diminishes when applied to solving the same problems using other programming languages [19]. This capability gap limits the universal applicability of these powerful LLMs and hinders developers working in diverse programming language ecosystems.

Consequently, it is crucial to enhance LLMs' proficiency across a wider range of programming languages to match their performance in widely used ones [26, 19]. We address this challenge by training LLMs on code translation tasks, requiring them to translate code from one programming

---

*Equal Contribution

language to another accurately. Intuitively, if LLMs can accurately translate Python code into other programming languages, they can achieve comparable performance in Python code generation tasks in those programming languages. Moreover, focusing on code translation facilitates the LLM's understanding of inter-language similarities and differences and allows it to learn from exemplary code structures.

Recently, on-policy RL algorithms [24, 1, 25, 14, 9] utilizing rule-based rewards have demonstrated significant performance in code generation tasks [7, 6]. However, the inherent coarse-grained nature of these rule-based rewards, lacking process-level supervision, limits their effectiveness in guiding LLMs to recognize the functional equivalence of intermediate code components during training. Implementing fine-grained process-level rewards is challenging, particularly given that multiple valid implementations may exist for the same code functionality. Given that leveraging preference data can guide LLM behavior towards desired outcomes, preference optimization [21, 3] offers an opportunity to incorporate process-level constraints during training.

Functional equivalence is crucial during the training process for code translation tasks. High-level programming languages can be too abstract for effective functional understanding by LLMs, especially for less common ones [26]. Consequently, interlingual intermediate representations (IRs) can be leveraged to facilitate cross-language transfer from high- to low(er)-resource programming languages [19, 10, 26]. Compiler IRs are agnostic to the source programming language and target execution platform, providing a method to align constructs from different programming languages semantically [5]. Therefore, we construct groups of IRs for the preference optimization process. By guiding the LLM to distinguish IRs equivalent to the source code from inequivalent ones and utilizing signals about the mutual equivalence between IRs within the group, the LLM can capture fine-grained information regarding code functionality.

This study introduces OORL, a novel RL framework integrating on-policy and off-policy strategies for training. First, we employ on-policy RL with a binary, rule-based reward signal. This component ensures the fundamental correctness of the code translation, verifying that the generated code adheres to formatting requirements, compiles successfully, and passes predefined unit tests. This provides a strong, unambiguous signal for task completion. Second, we introduce group equivalent preference optimization (GEPO) to incorporate finer-grained quality distinctions and address the unique nature of code generation, where multiple equivalences can exist. GEPO is a novel preference optimization method that extends beyond traditional pairwise comparisons. It leverages preference data comparing groups of "winner" (equivalent IRs) and "loser" (inequivalent IRs) responses. Crucially, GEPO explicitly models the concept of functional equivalence within the winner group, guiding the model to recognize that multiple distinct IRs can be equally valid. This allows the model to learn nuanced aspects of code functionality across multiple programming languages.

Our contributions can be summarized as follows:

- Introduce OORL, a novel RL framework for training with code translation tasks to enhance LLMs' understanding across multiple programming languages.

- Develop GEPO, a novel preference optimization method that extends beyond pairwise comparisons by explicitly modeling equivalence within IRs groups.

- Conduct extensive evaluations demonstrating the superior performance of our methods across various code benchmarks using multiple programming languages.

## 2  Preliminaries

### 2.1  RL with an Explicit Reward Function

RL [24, 7] has been widely used in preference optimization for LLMs. Let $\pi_{\text{ref}}$ denote an LLM obtained after supervised fine-tuning (SFT). RL optimizes the policy model $\pi_\theta$, typically initialized from $\pi_{\text{ref}}$ with parameters $\theta$, by maximizing an expected reward signal. This requires defining a reward function $R(x, y)$ that assigns a scalar score to a complete response $y$ given the prompt $x$. This score reflects the accuracy, quality, or alignment of the response with human preferences.

Once the reward function $R(x, y)$ is established, an RL algorithm can be employed to optimize the policy model $\pi_\theta$. The objective is typically to maximize the expected reward, often incorporating a KL divergence penalty term to prevent $\pi_\theta$ from deviating too far from the initial reference policy

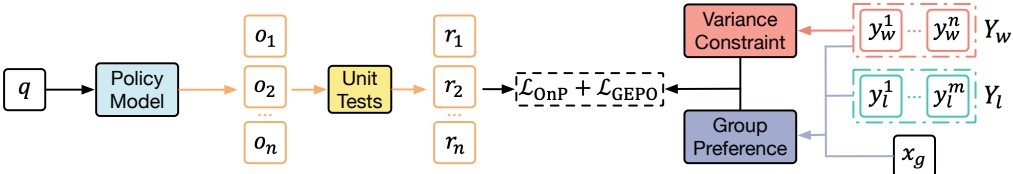

Figure 1: Overview of the OORL integrating on-policy RL with the rule-based reward and on-policy preference optimization (GEPO).

model $\pi_{ref}$, thereby maintaining generative capabilities and diversity:

$$\max_{\theta} \mathbb{E}_{x \sim D_{\text{prompt}}} \mathbb{E}_{y \sim \pi_\theta(\cdot|x)}[R(x, y) - \beta \mathbb{D}_{\text{KL}}(\pi_\theta(\cdot|x)||\pi_{\text{ref}}(\cdot|x))], \quad (1)$$

where $D_{\text{prompt}}$ is a dataset of prompts and $\beta$ is a hyperparameter controlling the KL penalty.

## 2.2 Direct Preference Optimization

Direct Preference Optimization (DPO) [21] is an alternative approach that leverages the preference dataset $D_{\text{pref}}$ to directly optimize the policy model $\pi_\theta$ without the need for explicitly training a separate reward model. DPO establishes a direct link between preference probabilities and policy probabilities, deriving an objective equivalent to optimizing against a learned reward model within the RLHF framework. Specifically, DPO assumes an implicit reward function $r_\phi(x, y)$ exists such that the human preference data can be modeled via the Bradley-Terry model:

$$P(y_w \succ y_l|x) = \sigma(r_\phi(x, y_w) - r_\phi(x, y_l)), \quad (2)$$

where $\succ$ denotes "is preferred over". Theoretical derivation shows that the RL objective of aligning with this implicit reward (including the KL penalty) can be transformed into a maximum likelihood objective computed directly on the preference data. The DPO loss function is defined as:

$$\mathcal{L}_{\text{DPO}}(\theta; \pi_{\text{ref}}) = -\mathbb{E}_{(x, y_w, y_l) \sim D_{\text{pref}}}[\log \sigma(\beta \log \frac{\pi_\theta(y_w|x)}{\pi_{\text{ref}}(y_w|x)} - \beta \log \frac{\pi_\theta(y_l|x)}{\pi_{\text{ref}}(y_l|x)})], \quad (3)$$

where $\pi_\theta$ is the policy model being optimized, $\pi_{\text{ref}}$ is the reference policy model, and $\beta$ is a scaling factor for the implicit reward.

## 3 Methods

Integrating on-policy (Section 3.1) and off-policy (Section 3.2) strategies, as illustrated in Figure 1, we introduce OORL (Section 3.3), a novel RL framework for multi-programming language understanding.

### 3.1 On-policy RL for Code Translation

Although LLMs achieve remarkable performance in code generation tasks, a significant performance disparity persists between high-resource programming languages (e.g., Python, C++) and others. To address this capability gap, we leverage the code translation task to train LLMs, thereby facilitating the transfer of coding proficiency across diverse programming languages. For example, if LLMs can accurately translate Python code into other programming languages, they can achieve comparable performance in Python code generation tasks while using those programming languages.

**RL with Binary Code Translation Reward**. Recently, on-policy RL algorithms utilizing rule-based rewards have demonstrated significant performance in code generation tasks [7, 6]. Consequently, we apply on-policy RL to training with code translation tasks. Let's first define the RL setup for the code translation task: (1) **State** ($s$): The state typically consists of the input $q$ with source code and potentially the sequence of tokens generated so far $o_{<t}$ with translated code; (2) **Action** ($a$): The action corresponds to the generation of the next token $o_t$ by the policy model; (3) **Policy** ($\pi_\theta$): The policy model $\pi_\theta(a|s)$ is the LLM being optimized, parameterized by $\theta$. It's typically initialized from

```c
int sum_sequential_integers(int count) {
    int current_sum = 0;
    for (int i = 0; i < count; ++i) {
        current_sum += i;
    }
    return current_sum;
}
```

```llvm
define dso_local i32 @sum_sequential_integers(i32 noundef
%count) local_unnamed_addr {
entry:
    %smax = tail call i32 @llvm.smax.i32(i32 %count, i32 0)
    %0 = zext nneg i32 %smax to i33
    %1 = add nsw i32 %smax, -1
    %2 = zext i32 %1 to i33
    %3 = mul i33 %0, %2
    %4 = lshr i33 %3, 1
    %5 = trunc nuw i33 %4 to i32
    ret i32 %5
}
declare i32 @llvm.smax.i32(i32, i32) #1
```

```llvm
define dso_local i32 @sum_sequential_integers(i32 noundef
%count) local_unnamed_addr {
entry:
    %cmp4 = icmp sgt i32 %count, 0
    br i1 %cmp4, label %for.body.preheader, label
%for.cond.cleanup
for.body.preheader:
    %0 = add nsw i32 %count, -1
    %1 = zext nneg i32 %0 to i33
    %2 = add nsw i32 %count, -2
    %3 = zext i32 %2 to i33
    %4 = mul i33 %1, %3
    %5 = lshr i33 %4, 1
    %6 = trunc nuw i33 %5 to i32
    %7 = add i32 %count, %6
    %8 = add i32 %7, -1
    br label %for.cond.cleanup
for.cond.cleanup:
    %current_sum.0.lcssa = phi i32 [ 0, %entry ], [ %8,
%for.body.preheader ]
    ret i32 %current_sum.0.lcssa
}
```

Source code $x_g$ and corresponding function equivalenct IRs group $Y_w$

```llvm
define dso_local i32 @sum_sequential_integers(i32 noundef
%count) local_unnamed_addr {
entry:
    %smax = tail call i32 @llvm.smax.i32(i32 %count, i32 0)
    %0 = add nsw i32 %smax, -1
    %1 = zext i32 %0 to i33
    %2 = mul i33 %0, %1
    %3 = lshr i33 %2, 1
    %4 = trunc nuw i33 %3 to i32
    ret i32 %4
}
declare i32 @llvm.smax.i32(i32, i32) #1
```

```llvm
define dso_local i32 @sum_sequential_integers(i32 noundef
%count) local_unnamed_addr {
entry:
    %smax = tail call i32 @llvm.smax.i32(i32 %count, i32 0)
    %0 = zext nneg i32 %smax to i33
    %1 = add nsw i32 %smax, -1
    %2 = zext i32 %1 to i33
    %3 = mul i33 %0, %2
    %4 = lshr i33 %3, 1
    ret i32 %4
}
declare i32 @llvm.smax.i32(i32, i32) #1
```

Inequivalent IRs group $Y_l$, augmented from equivalenct IRs group $Y_w$

Figure 2: Overview of the equivalent and equivalent IRs groups for the training process with GEPO. The equivalent IRs group is constructed from different LLVM optimization levels (e.g., -Oz, -O3) of the source code. The inequivalent IRs group is augmented from the equivalent IRs group.

the SFT reference model $\pi_{\text{ref}}$. During the on-policy RL process, given the query $q$ and the trajectory $o$, we employ a binary reward function, $R_{\text{rule}}(q, o)$, assigned upon completion of the generation $o$:

$$R_{\text{rule}}(q, o) = \begin{cases} 1, & \text{if } o \text{ represents a successful code translation from } q, \\ 0, & \text{otherwise.} \end{cases} \tag{4}$$

Specifically, successful translation demonstrates that $o$ must adhere to the standard format specified in $q$ and the translated code extracted from $o$ can be successfully compiled and pass unit tests. The policy model $\pi_\theta$ is optimized using an on-policy RL algorithm (e.g., PPO, GRPO) to optimize Equation (1). The on-policy RL algorithm operates by iteratively sampling trajectories $(q, o)$ from the current policy model $\pi_\theta$. For each action in each sampled trajectory, it estimates the advantage $A_t$, which quantifies how much better the received reward $R_t$ is compared to an expected baseline. The policy parameters $\theta$ are then updated using a clipped objective function designed to maximize the expected advantage while limiting the change in the $\pi_\theta$ at each step. The on-policy RL loss can be defined as:

$$\mathcal{L}_{\text{OnP}} = -\mathbb{E}_{(q,o)\sim\pi_\theta}[\min(r_t(\pi_\theta)A_t, \text{clip}(r_t(\pi_\theta), 1 - \epsilon, 1 + \epsilon)A_t)], \tag{5}$$

where $r_t(\pi_\theta) = \frac{\pi_\theta(a|s)}{\pi_{\theta_{\text{old}}}(a|s)}$ is the probability ratio between new and old policies. $\epsilon$ is a hyperparameter defining the clipping range. This rule-based RL component, focused on maximizing the binary success signal defined in Equation (4). It provides a strong, unambiguous signal for task completion, which is complemented by the finer-grained preference information incorporated through the GEPO loss, as described in the subsequent section.

## 3.2 Group Equivalent Preference Optimization with IRs

The inherent coarse-grained nature of rule-based rewards, which lack process-level supervision, limits their effectiveness in guiding LLMs to recognize the functional equivalence of intermediate code components. Implementing fine-grained process-level rewards is challenging, particularly as multiple valid implementations can exist for the same code functionality. Given that leveraging preference data can effectively guide LLM behavior towards desired outcomes, preference optimization [21, 3] offers a promising avenue to incorporate process-level constraints during training.

Functional equivalence is crucial in code generation scenarios. However, the functions implemented with high-level programming languages can be too abstract for LLMs to understand, especially

for less common ones. In contrast, compiler IRs expose detailed and low-level operators that are significantly easier for LLMs to understand [26]. Consequently, we propose performing preference optimization using IR translation tasks, which require LLMs to translate code written in high-level programming languages into IRs. This process facilitates the LLM's understanding of the code functionality expressed in the high-level languages. Furthermore, equivalent IRs resulting from different compiler optimization stages can guide LLMs in discerning the equivalent functionality of different code implementations.

However, traditional preference optimization methods [21, 3] typically rely on pairwise comparisons, which are insufficient in this context. Specifically, methods like DPO [21] do not account for the equivalence among IRs. Therefore, we introduce Group Equivalent Preference Optimization (GEPO) and construct groups $D_{\text{gpref}}$ with equivalent and inequivalent IRs for the preference optimization process. GEPO compares groups of IR responses and specifically incorporates the concept of equivalence within the group of preferred ("winner") IR responses. By guiding the LLM to distinguish IRs equivalent to the source code from inequivalent ones and utilizing signals about the mutual equivalence between IRs within a group, GEPO enables the LLM to capture detailed information regarding code functionality.

For each prompt $x_g$ with source code in $D_{\text{gpref}}$ for GEPO, there exists a group of "winner" responses $Y_w = \{y_w^1, y_w^2, \cdots, y_w^n\}$, a group of "loser" responses $Y_l = \{y_l^1, y_l^2, \cdots, y_l^m\}$ and $y_g \in (Y_w \cup Y_l)$. Specifically, $Y_w$ represents $n$ functionally equivalent IRs while $Y_l$ represents $m$ inequivalent IRs. Following [21], we assume an implicit reward function $r_\phi(x_g, y_g)$ underlies the preferences. Instead of comparing the individual winner IR response and the loser IR response, GEPO compares the average reward of $Y_w$ to the average reward of $Y_l$. Furthermore, it explicitly enforces equivalence within $Y_w$ by constraining the variance of their rewards. The initial optimization problem for GEPO, formulated in terms of the implicit reward $r_\phi$, can be defined as:

$$\min_\phi \mathcal{L}_{\text{GEPO}} = -\mathbb{E}_{(x_g, Y_w, Y_l) \sim D_{\text{gpref}}}[\log \sigma(\frac{1}{n} \sum_{i=1}^n r_\phi(x_g, y_w^i) - \frac{1}{m} \sum_{k=1}^m r_\phi(x_g, y_l^k))]$$

$$\text{s.t. } \mathbb{E}_{(x_g, Y_w) \sim D_{\text{gpref}}}[\text{Var}(r_\phi(x_g, Y_w))] < \epsilon,$$

$$\text{Var}(r_\phi(x_g, Y_w)) = \frac{1}{n-1} \sum_{i=1}^n (r_\phi(x_g, y_w^i) - \frac{1}{n} \sum_{j=1}^n r_\phi(x_g, y_w^j))^2. \tag{6}$$

The primary objective term maximizes the probability that the average winner reward exceeds the average loser reward, modeled using the sigmoid function. The constraint bounds the variance of rewards within the winner group $Y_w$ by a small threshold $\epsilon$, encouraging $r_\phi(x_g, y_w^i) \approx r_\phi(x_g, y_w^j)$ for all $y_w^i, y_w^j \in Y_w$.

According to Appendix A, we can establish a direct relationship between the implicit reward function $r_\phi$ and the optimal policy $\pi_\theta$ being optimized, relative to the reference policy model $\pi_{\text{ref}}$. Accordingly, the optimal policy model $\pi^*$ that maximizes Equation (1) can be given by $\pi^*(y_g|x_g) \propto \pi_{\text{ref}}(y_g|x_g) \exp(\frac{1}{\beta} r_\phi(x_g, y_g))$. This implies that the reward function can be expressed in terms of the policy probabilities, up to a partition function $Z(x_g)$, as follows:

$$r_\phi(x_g, y_g) = \beta \log \frac{\pi_\theta(y_g|x_g)}{\pi_{ref}(y_g|x_g)} + \beta \log Z(x_g), \tag{7}$$

where $\pi_\theta$ is the policy model to be optimized. Substituting this expression for $r_\phi(x_g, y_g)$ into Equation (6), the partition function terms $\beta \log Z(x_g)$ cancel out within the reward differences and the variance calculation. This yields an objective function that depends on the policy model $\pi_\theta$, the reference policy model $\pi_{\text{ref}}$, and the preference data $(x_g, Y_w, Y_l)$, which can be formulated as follows:

$$\min_\theta \mathcal{L}'_{\text{GEPO}} = -\mathbb{E}_{(x_g, Y_w, Y_l) \sim D_{\text{gpref}}}[\log \sigma(\frac{\beta}{n} \sum_{i=1}^n \hat{r}_\theta(x_g, y_w^i) - \frac{\beta}{m} \sum_{k=1}^m \hat{r}_\theta(x_g, y_l^k))]$$

$$\text{s.t. } \mathbb{E}_{(x_g, Y_w) \sim D_{\text{gpref}}}[\frac{\beta^2}{n-1} \sum_{i=1}^n (\hat{r}_\theta(x_g, y_w^i) - \frac{1}{n} \sum_{j=1}^n \hat{r}_\theta(x_g, y_w^j))^2] < \epsilon',$$

$$\hat{r}_\theta(x_g, y_g) = \log \frac{\pi_\theta(y_g|x_g)}{\pi_{\text{ref}}(y_g|x_g)}. \tag{8}$$

Here, $\hat{r}_\theta(x_g, y_g)$ represents the log-probability ratio proportional to the implicit reward score. The constraint now applies to the variance of these log-probability ratios within the winner group. To solve this constrained optimization problem, we introduce a Lagrange multiplier $\lambda \geq 0$ for the variance constraint, transforming it into an unconstrained objective. The GEPO loss can be formulated as:

$$\mathcal{L}_{\text{GEPO}} = -\mathbb{E}_{(x_g, Y_w, Y_l) \sim D_{\text{gpref}}}[\log \sigma(\beta(\frac{1}{n}\sum_{i=1}^{n}\hat{r}_\theta(x_g, y_w^i) - \frac{1}{m}\sum_{k=1}^{m}\hat{r}_\theta(x_g, y_l^k)))]$$

$$+ \lambda \mathbb{E}_{(x_g, Y_w) \sim D_{\text{gpref}}}[\frac{\beta^2}{n-1}\sum_{i=1}^{n}(\hat{r}_\theta(x_g, y_w^i) - \frac{1}{n}\sum_{j=1}^{n}\hat{r}_\theta(x_g, y_w^j))^2]. \tag{9}$$

Finally, $\mathcal{L}_{\text{GEPO}}$ consists of two terms. The first term is analogous to the DPO loss but operates on the average log-probability ratios of the winner and loser groups, pushing the model to prefer winners over losers on average. The second term, weighted by the hyperparameter $\lambda$, explicitly penalizes the variance of the log-probability ratios within the winner group $Y_w$. Minimizing this term encourages $r_\phi(x_g, y_g)$ to be similar for all winners $y_w \in Y_w$, thereby enforcing the desired equivalence. The hyperparameter $\lambda = 1$ controls the strength of this equivalence regularization relative to the main preference objective. By minimizing $\mathcal{L}_{\text{GEPO}}$, we directly optimize the policy model $\pi_\theta$ to enhance the likelihood of high-quality, functionally equivalent code translation ($Y_w$) compared to undesired ones ($Y_l$), while simultaneously promoting diversity and consistency among the preferred solutions.

### 3.3 OORL

Building upon the on-policy RL strategy and GEPO, we develop OORL, a novel RL framework for training with code translation tasks, as shown in Figure 1. Specifically, following the on-policy nature used for the RL part, we periodically sample trajectories $(q, o)$ using the current policy model $\pi_\theta$ and compute the binary reward $R_{\text{rule}}(q, o)$. This allows the estimation of advantages $A_t$ and the computation of the on-policy RL objective $\mathcal{L}_{\text{OnP}}(\theta)$, which drives the policy model towards successful task completion according to the defined rules. In parallel, we utilize the static preference dataset $D_{\text{grefs}} = \{(x_g, Y_w, Y_l)\}$ containing groups of winner and loser responses to calculate $\mathcal{L}_{\text{GEPO}}(\pi_\theta; \pi_{\text{ref}})$, which guides the policy model to align with fine-grained function equivalences among desirable IRs.

The parameters $\theta$ of $\pi_\theta$ are updated using a combined signal derived from both objectives. A conceptual representation of the combined objective to be minimized during policy updates can be expressed as:

$$\mathcal{L} = w_{\text{rl}} \cdot \mathcal{L}_{\text{OnP}} + w_{\text{gepo}} \cdot \mathcal{L}_{\text{GEPO}}, \tag{10}$$

where the non-negative $w_{\text{rl}}$ and $w_{\text{gepo}}$ serve as weights that balance the contribution of the rule-based task completion signal from RL against the nuanced quality and equivalence signal from GEPO. Specifically, we set $w_{\text{rl}} = 1$ and $w_{\text{gepo}} = 0.01$ for the training process with OORL.

## 4 Experiments

### 4.1 Experiment Settings

**Training Data**. The training dataset for OORL is constructed separately. For on-policy RL, we construct 2400 code translation problems with unit tests from the SYNTHETIC-1 [17] and APPS [8] datasets. Specifically, this batch of training data includes translation from Python and C++ as source codes to Python, C++, Java, PHP, Bash, and JavaScript as target codes. Meanwhile, GEPO is performed with 9600 curated C-to-IR groups from the SLTrans dataset [19].

**Implementation Details**. In this paper, we implement our OORL framework based on Qwen3-8B [20]. Specifically, we employ REINFORECE++ [1, 9] as the on-policy RL algorithm with OORL. The training process entailed using a cosine learning rate schedule with a warm-up ratio of 0.03, and the AdamW [15] optimizer with a learning rate of $5 \times 10^{-7}$, no weight decay, a batch size of 8, and a sequence length of 8192 tokens. The models underwent instruction tuning for one epoch using DeepSpeed-Zero stage2 with offload [22] on 4 A100 GPUs, each with 80G memory.

**Baselines**. Our Qwen3-8B-OORL, trained with the OORL framework, is compared against existing LLMs with comparable parameters. These include DS-Coder-V2-Lite-Inst [27], Qwen2.5-Coder-7B-

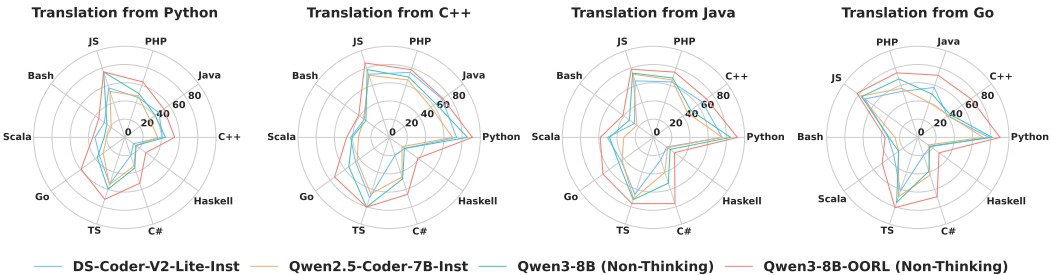

Figure 3: Code Translation Performance of LLMs on CrossPLEval.

Table 1: Statistics in the CrossPLEval Benchmark.

| Programming Language | # Solution Files | Avg. Lines of Code (LoC) | Proportion of Files (%) |
|---|---|---|---|
| MultiPL-E(Python) | – | 7.81 | – |
| Python | 119×10 | 19.94 | 30.36 |
| C++ | 92×10 | 35.49 | 23.47 |
| Go | 92×10 | 47.60 | 23.47 |
| Java | 89×10 | 39.72 | 22.70 |
| Total / Weighted Avg. | 392×10 | 34.57 | 100.00 |

Avg. LoC for the total is a weighted average. Proportions are based on the total of 3920 files.

Inst [11], and Qwen3-8B [20]. Specifically, the experiments with Qwen3-8B are conducted using its non-thinking mode.

**Evaluation Benchmarks**. To comprehensively evaluate proficiency in multi-programming languages understanding, we utilize the MultiPL-E [4] benchmark and select eight mainstream languages for evaluation, including Python, C++, Java, PHP, TypeScript, C#, Bash, and JavaScript. In addition, we develop CrossPLEval, a new multilingual code translation benchmark, for further evaluation. CrossPLEval comprises 119 programming problems carefully selected from the TACO dataset [13], with details illustrated in Section 4.2. For evaluation, we use the pass@1 rate as the metric for MultiPL-E and CrossPLEval benchmarks.

## 4.2 CrossPLEval

CrossPLEval is designed to assess the ability of models to translate a canonical solution of a problem from a source language to a target language. Specifically, the model receives the function declaration and the canonical solution in the source language (e.g., Python) as input and is tasked with generating the corresponding solution in the target language. To facilitate evaluation, the target language function declaration is also provided, which helps constrain function names and variable types.

The 119 core problems in CrossPLEval originate from the TACO dataset [13], initially defined in Python. Recognizing the non-trivial nature of developing high-quality, semantically aligned test data across languages, we undertook meticulous manual processing for these problems to ensure data diversity and quality. This process involved two main efforts. First, concerning canonical solutions, while reference Python solutions were available for all 119 problems, we manually rewrote these into equivalent implementations in other mainstream programming languages. This yielded 89 solutions for Java, 92 for Go, and 92 for C++, resulting in 273 high-quality translation task instances from Python to these three target languages. Second, to ensure rigorous evaluation of functional correctness, we manually rewrote the original test cases for each problem into ten different programming languages, including C++, Java, PHP, JavaScript, Bash, Scala, Go, TypeScript, C#, and Haskell. These multilingual test cases enable verification of translation accuracy across a broad linguistic spectrum. Through this construction process, the CrossPLEval benchmark comprises 3920 independent coding problems, each of which includes at least 5 unit tests, thereby providing a solid foundation for evaluation. The details are described in Table 1.

Table 2: Performance of different LLMs on MultiPL-E [4].

| | Python | C++ | Java | PHP | Bash | JS | TS | C# | Avg. |
|---|---|---|---|---|---|---|---|---|---|
| DS-Coder-V2-Lite-Inst | 81.10 | 32.91 | 68.35 | 72.67 | 19.62 | 81.36 | 83.33 | 41.13 | 60.06 |
| Qwen2.5-Coder-7B-Inst | 88.40 | 57.14 | 70.88 | 73.91 | 43.67 | 78.88 | 84.27 | 52.53 | 68.71 |
| Qwen3-8B | 82.60 | 71.42 | 70.25 | 50.31 | 36.07 | 83.22 | 84.27 | 42.41 | 65.07 |
| **Qwen3-8B-OORL** | **90.06** | **83.23** | **83.54** | **78.26** | **46.20** | **89.44** | **85.53** | **54.22** | **76.31** |

Table 3: Results of code translation compared with other LLMs on CrossPLEval.

| | Model | Target Language | | | | | | | | | | | |
|---|---|---|---|---|---|---|---|---|---|---|---|---|---|
| | | Python | C++ | Java | PHP | JS | Bash | Scala | Go | TS | C# | Haskell | Avg. |
| Py | DS-Coder-V2-Lite-Inst | - | 41.17 | 46.21 | 46.21 | 57.14 | 27.73 | 32.77 | 34.45 | 53.78 | 21.00 | 11.76 | 37.22 |
| | Qwen2.5-Coder-7B-Inst | - | 36.13 | 33.61 | 47.89 | 52.94 | 18.54 | 20.16 | 29.41 | 54.62 | 32.77 | 15.12 | 34.12 |
| | Qwen3-8B | - | 44.53 | 41.17 | 50.42 | 75.54 | 24.36 | 22.68 | 37.25 | 59.66 | 35.29 | 15.13 | 40.60 |
| | **Qwen3-8B-OORL** | - | **54.62** | **52.94** | **63.94** | **75.63** | **35.29** | **35.29** | **59.32** | **71.42** | **52.94** | **28.48** | **52.99** |
| C++ | DS-Coder-V2-Lite-Inst | 73.91 | - | 71.73 | 75.00 | 72.82 | 33.69 | 42.39 | 45.65 | 69.56 | 20.65 | 20.65 | 52.61 |
| | Qwen2.5-Coder-7B-Inst | 69.56 | - | 56.52 | 66.30 | 71.73 | 34.78 | 33.69 | 43.47 | 65.21 | 45.65 | 17.39 | 50.43 |
| | Qwen3-8B | 85.86 | - | 63.04 | 69.56 | 78.26 | 31.52 | 40.65 | 54.71 | 80.43 | 47.82 | 20.65 | 57.25 |
| | **Qwen3-8B-OORL** | **91.30** | - | **72.82** | **78.15** | **85.86** | **41.30** | **46.73** | **74.02** | **80.43** | **65.65** | **38.91** | **67.52** |
| Java | DS-Coder-V2-Lite-Inst | 75.28 | 66.29 | - | 64.04 | 65.16 | 25.84 | 50.56 | 43.82 | 65.82 | 25.84 | 17.97 | 50.06 |
| | Qwen2.5-Coder-7B-Inst | 75.28 | 46.06 | - | 66.29 | 73.03 | 22.47 | 32.58 | 42.69 | 70.78 | 40.44 | 19.10 | 48.87 |
| | Qwen3-8B | 84.64 | 50.56 | - | 68.53 | 74.15 | 32.58 | 48.31 | 47.19 | 71.91 | 52.80 | 22.47 | 55.31 |
| | **Qwen3-8B-OORL** | **92.13** | **71.91** | - | **75.28** | **78.65** | **41.34** | **58.42** | **68.53** | **76.40** | **76.40** | **28.53** | **66.76** |
| Go | DS-Coder-V2-Lite-Inst | 78.26 | 40.21 | 57.60 | 53.26 | 71.73 | 32.60 | 32.60 | - | 61.95 | 20.65 | 17.39 | 46.63 |
| | Qwen2.5-Coder-7B-Inst | 71.73 | 41.30 | 40.21 | 56.52 | 77.17 | 25.00 | 26.08 | - | 68.47 | 43.47 | 14.13 | 46.41 |
| | Qwen3-8B | 81.52 | 43.47 | 50.00 | 67.39 | 76.08 | 31.52 | 26.08 | - | 75.00 | 39.13 | 16.30 | 50.65 |
| | **Qwen3-8B-OORL** | **90.21** | **72.06** | **71.73** | **74.45** | **82.39** | **35.86** | **48.26** | - | **81.08** | **68.47** | **28.80** | **65.33** |

To illustrate the difficulty of CrossPLEval across different languages, we visualize the code translation performance in Figure 3. Current SOTA code LLMs with around 8B parameters achieve a low score on average performance across multiple programming languages.

## 4.3 Excellent Multi-Programming Language Understanding

To evaluate proficiency in understanding and generating code across multiple programming languages, we conduct evaluations using the established MultiPL-E [4] benchmark and our newly introduced CrossPLEval benchmark. Besides Python, C++, Java, PHP, Bash, and JavaScript, we also include Scala, Go, TypeScript, C#, and Haskell in our evaluation, noting that the latter five languages are not included in the training dataset. Moreover, Bash, Scala, and Haskell are particularly noteworthy as low-resource programming languages.

Table 2 presents the code generation performance on MultiPL-E, while Table 3 and Figure 3 show code translation results on CrossPLEval. Across both benchmarks and all tested models, our Qwen3-8B-OORL model consistently demonstrates superior performance. On MultiPL-E, it achieves the highest average score of 76.31, significantly outperforming the second highest Qwen2.5-Coder-7B-Inst with a significant margin. This strong performance extends across all eight languages evaluated in MultiPL-E, which are generally well-represented in large code corpora. This highlights the excellent ability to generate correct code from natural language instructions in these programming languages.

Furthermore, the CrossPLEval results demonstrate the code translation capabilities and, critically, its generalization ability. Qwen3-8B-OORL achieves the highest average translation scores across all source languages evaluated. The CrossPLEval benchmark is particularly valuable for evaluating performance on languages potentially less represented in training data, such as Scala, Go, TypeScript, C#, and the low-resource Haskell. Although absolute translation scores into target languages like Bash, Scala, and Haskell are lower compared to more common targets, our Qwen3-8B-OORL exhibits significant relative improvements over the baselines on these more challenging targets. For instance, when translating from Python to Haskell, Qwen3-8B-OORL scores 28.48, representing a substantial

Table 4: Ablation studies on various benchmarks using OORL.

| | On-Policy RL Strategy | Off-Policy RL Strategy | MultiPL-E | CrossPLEval | | | | |
|---|---|---|---|---|---|---|---|---|
| | | | | Python | C++ | Java | Go | Avg. |
| Qwen3-8B | - | - | 65.06 | 40.60 | 57.25 | 55.32 | 50.65 | 50.96 |
| | REINFORCE++ | - | 73.60 | 47.56 | 63.26 | 62.13 | 60.87 | 58.46 |
| | REINFORCE++ | DPO | 73.48 | 50.93 | 62.93 | 62.80 | 58.15 | 58.70 |
| | REINFORCE++ | GEPO | **76.31** | **52.99** | **67.52** | **66.76** | **65.33** | **63.15** |

gain over the Qwen3-8B. These results underscore the effective generalization beyond its direct training data and its ability to provide significant performance boosts for tasks involving low-resource programming languages.

The evaluation results demonstrate that Qwen3-8B-OORL not only excels in core code generation tasks on well-represented languages but also effectively translates and generalizes to improve performance on untrained and low-resource programming languages, marking a notable advancement in multilingual code understanding and generation.

## 4.4 Ablation Studies

To evaluate the individual contribution of each key component within our OORL framework, we conducted thorough ablation studies. The results of these studies are detailed in Table 4. The investigation systematically deconstructed OORL by starting with the Qwen3-8B model and incrementally adding the on-policy RL strategy (REINFORCE++), followed by a comparison of different off-policy preference optimization strategies, specifically DPO [21] and our proposed GEPO.

**On-Policy RL Strategy**. As illustrated in Table 4, training with code translation tasks using the REINFORCE++ [9, 1] algorithm increases the MultiPL-E score to 73.60 and the average CrossPL-Eval score to 58.46, demonstrating a significant performance improvement. This substantial gain demonstrates the effectiveness of on-policy RL, which directly refines multi-programming language understanding of LLMs through the rule-based reward derived from unit tests.

**Off-Policy RL Strategy**. As shown in Table 4, the combination of REINFORCE++ [9, 1] and DPO [21] only offers a marginal improvement on CrossPLEval. The effectiveness of DPO in code understanding is found to be limited. This limitation stems from the difficulty standard preference optimization methods face in fully understanding the functional equivalence between different code implementations. In contrast, our GEPO strategy is specifically designed to utilize functional equivalence information from IRs. It's worth noting that integrating GEPO with REINFORCE++ achieved significantly higher performance, reaching 76.31 on MultiPL-E and a notable average of 63.15 on CrossPLEval. This demonstrates the ability of our GEPO to leverage functional equivalence, providing substantial additional performance gains over standard off-policy methods in the context of multi-programming language understanding.

## 5 Related Works

### 5.1 RL for LLMs

RL [24, 25, 21, 3] has emerged as a prominent paradigm for aligning LLMs with desired behaviors and preferences. Investigations in this area have explored both on-policy and off-policy RL algorithms. On-policy methods [24, 1, 25, 14, 9] have received considerable attention due to their inherent stability and effectiveness in directly optimizing the policy using data collected under the current policy. For instance, PPO [24] has been widely adopted for fine-tuning LLMs based on human preference data. Similarly, REINFORCE-based methods, such as ReMax [14], RLOO [1], and GRPO [25], have been extensively employed in extending mathematical and general reasoning abilities, often utilizing rule-based rewards exclusively. Concurrently, off-policy algorithms offer potential advantages in terms of sample efficiency by allowing the agent to learn from prior experiences or data generated by different policies. DPO [21] and IPO [3] have been proposed as more stable and efficient RL techniques that implicitly learn a reward function from preference data and directly optimize the policy. These methods can be conceptualized as implicitly performing off-policy evaluation and

optimization. Building upon the successes of both on-policy and off-policy RL, we introduce OORL, a novel RL framework that integrates on-policy and off-policy strategies for training. We also incorporate GEPO, a novel off-policy method that leverages the concept of functional equivalence, to provide a more nuanced and effective approach for training LLMs in code translation tasks.

## 5.2 Code Understanding with IRs

Intermediate Representations (IRs) have been increasingly leveraged in recent work to enhance LLMs for code generation tasks. The language-agnostic nature of IRs allows LLMs to abstract away from specific syntax and focus more effectively on program logic. For instance, to improve multilingual code generation and cross-lingual transfer, IRCoder [19] employs continued pre-training of LLMs on SLTrans, a parallel dataset of source code and its corresponding LLVM IR, thereby aligning code and IR semantics. Similarly, Transcoder-IR[26] also utilizes LLVM IR, augmenting LLM training by exploring it as a pivot language. Acknowledging the potential complexities of standard compiler IRs for translation tasks, CoDist [10] proposes a custom, distilled code as a simplified intermediate representation and translation pivot. Furthermore, for compiler-specific applications, LLM Compiler [5] is specialized through extensive further pre-training of Code LLMs directly on large corpora of LLVM IR and assembly code. This specialization aims to enhance performance on tasks such as code optimization and disassembly. In this paper, we adopt a unique approach by leveraging LLVM IRs within our GEPO preference optimization process. We construct and compare groups of IRs derived from translated code. This method guides the LLM to discern fine-grained functional equivalence, which is crucial for effective cross-lingual code understanding.

# 6 Conclusion

In this paper, we introduce OORL, a novel RL framework that integrates both on-policy and off-policy strategies for training LLMs. Within the OORL framework, on-policy RL is applied during the code translation process, guided by a rule-based reward signal derived from unit tests. Complementing this coarse-grained rule-based reward, we propose Group Equivalent Preference Optimization (GEPO), a novel preference optimization method. GEPO specifically extends beyond traditional pairwise comparisons by explicitly modeling equivalence within groups of IRs. Extensive experiments demonstrate that our OORL framework achieves significant performance improvements on code benchmarks across multiple programming languages, highlighting its effectiveness for enhancing multilingual code understanding and generation. In general, this work presents a novel integrated RL approach that leverages functional equivalence and offers a new perspective for advancing LLMs' understanding capabilities across diverse programming languages.

# Limitations

A primary limitation of OORL stems from the group-based optimization mechanism in GEPO, which processes grouped preference responses for each source prompt. Compared to previous preference optimization methods [21, 3] that handle a pair of responses per step, GEPO manages larger groups of preference data, necessitating more GPU memory resources. This increased memory can be particularly pronounced with larger group sizes or longer response sequences. One potential strategy to mitigate this issue is to serialize the processing of individual responses within each preference group, which can reduce the number of padding tokens and thereby lower memory overhead. Although the serialization is beneficial for memory efficiency, it introduces an increase in training latency due to the sequential handling of responses previously processed in parallel within the group.

# Acknowledgement

This work is partially supported by The Research Grants Council of Hong Kong SAR (No. RFS2425-4S02 and No. CUHK14201624).

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

# A  Derivation of the Implicit Reward

This section details the derivation of the relationship between an implicit reward function $r_\phi(x_g, y_g)$ and the optimal policy $\pi_\theta(y_g|x_g)$, under the standard reinforcement learning objective with a KL-divergence penalty against a reference policy $\pi_{\text{ref}}(y_g|x_g)$. This formulation is foundational to methods like Direct Preference Optimization (DPO) and our proposed Group Equivalent Preference Optimization (GEPO).

Following [21], the objective is to find a policy $\pi_\theta$ that maximizes the expected reward $r_\phi(x_g, y_g)$ while remaining close to a reference policy $\pi_{\text{ref}}$, controlled by a coefficient $\beta$:

$$
\max_{\pi_\theta} \quad \mathbb{E}_{x_g \sim D, y_g \sim \pi_\theta(y_g|x_g)}[r_\phi(x_g, y_g)] - \beta \mathbb{D}_{\text{KL}}[\pi_\theta(y_g|x_g)||\pi_{\text{ref}}(y_g|x_g)]
$$

$$
= \min_{\pi_\theta} \quad \mathbb{E}_{x_g \sim D}\mathbb{E}_{y_g \sim \pi_\theta(y_g|x_g)}[\log \frac{\pi_\theta(y_g|x_g)}{\frac{1}{Z(x_g)}\pi_{\text{ref}}(y_g|x_g)\exp(\frac{1}{\beta}r_\phi(x_g, y_g))} - \log Z(x_g)]
$$

$$
= \min_{\pi_\theta} \quad \mathbb{E}_{x_g \sim D}[\mathbb{D}_{\text{KL}}(\pi_\theta(y_g|x_g)||\pi^*(y_g|x_g)) - \log Z(x_g)]. \tag{11}
$$

Here, $D$ represents the distribution of prompts $x_g$, and $\pi^*(y_g|x_g)$ is defined as the optimal policy distribution:

$$
\pi^*(y_g|x_g) = \frac{1}{Z(x_g)}\pi_{\text{ref}}(y_g|x_g)\exp(\frac{1}{\beta}r_\phi(x_g, y_g)). \tag{12}
$$

The term $Z(x_g)$ is the partition function, ensuring that $\pi^*(y_g|x_g)$ normalizes to a valid probability distribution over all possible responses $y_g$:

$$
Z(x_g) = \sum_{y_g} \pi_{\text{ref}}(y_g|x_g)\exp(\frac{1}{\beta}r_\phi(x_g, y_g)). \tag{13}
$$

The KL divergence term $\mathbb{D}_{\text{KL}}(\pi_\theta(y_g|x_g)||\pi^*(y_g|x_g))$ in Equation 11 is minimized (becomes zero) when the policy $\pi_\theta(y_g|x_g)$ is identical to $\pi^*(y_g|x_g)$. Therefore, the optimal policy $\pi_\theta$ that solves the optimization problem is:

$$
\pi_\theta(y_g|x_g) = \pi^*(y_g|x_g) = \frac{1}{Z(x_g)}\pi_{\text{ref}}(y_g|x_g)\exp(\frac{1}{\beta}r_\phi(x_g, y_g)). \tag{14}
$$

To derive the expression for the implicit reward $r_\phi(x_g, y_g)$, we can rearrange Equation 14:

$$
\frac{\pi_\theta(y_g|x_g)Z(x_g)}{\pi_{\text{ref}}(y_g|x_g)} = \exp(\frac{1}{\beta}r_\phi(x_g, y_g))
$$

$$
\log(\frac{\pi_\theta(y_g|x_g)Z(x_g)}{\pi_{\text{ref}}(y_g|x_g)}) = \frac{1}{\beta}r_\phi(x_g, y_g)
$$

$$
\beta(\log \frac{\pi_\theta(y_g|x_g)}{\pi_{\text{ref}}(y_g|x_g)} + \log Z(x_g)) = r_\phi(x_g, y_g).
$$

This gives us the final form for the implicit reward function:

$$
r_\phi(x_g, y_g) = \beta \log \frac{\pi_\theta(y_g|x_g)}{\pi_{\text{ref}}(y_g|x_g)} + \beta \log Z(x_g). \tag{15}
$$

This relationship (Equation 15) demonstrates that the implicit reward $r_\phi(x_g, y_g)$ is proportional to the log-probability ratio of the policy $\pi_\theta(y_g|x_g)$ with respect to the reference policy $\pi_{\text{ref}}(y_g|x_g)$, plus a term related to the partition function. This expression is leveraged in Section 3.2 (specifically, Equation 7) to transform the GEPO objective, which is initially defined in terms of $r_\phi(x_g, y_g)$, into an objective that directly depends on the policy probabilities $\pi_\theta(y_g|x_g)$ and $\pi_{\text{ref}}(y_g|x_g)$.

