# OpenReview forum: "On-Policy Optimization with Group Equivalent Preference for Multi-Programming Language Understanding"
_NeurIPS.cc/2025/Conference — NeurIPS 2025 poster_

### Official Review · Reviewer_w3YY · 2025-07-02

[review text omitted: it was posted to a different submission]

---

> ### Author Rebuttal · Authors · 2025-07-29
>
> We appreciate the **Reviewer w3YY**'s time and effort in reviewing our manuscript.
>
> We believe there might have been a mix-up with the review comments. Specifically, the comment appears to be unrelated to our submission. It seems to refer to a different paper.
> We understand that handling numerous submissions can be challenging, and such mix-ups can occasionally occur.
> We respectfully request your assistance and thanks you for your attention to this issue and your continuous support in this review process.

---

### Official Review · Reviewer_ms4a · 2025-07-02

**Clarity:** 3
**Significance:** 2
**Originality:** 3
**Rating:** 3
**Confidence:** 3

**Summary:**

This work aims to bridge the performance gap of large language models (LLMs) across different programming languages through code-to-code translation. The proposed approach, OORL, is a reinforcement learning (RL) framework that integrates on-policy fine-tuning using an enhanced REINFORCE algorithm (REINFORCE++) with off-policy preference learning. A key component of the method, GEPO, learns to rank groups of functionally equivalent compiler intermediate representations (IRs), enabling deep semantic alignment through preference optimization. The approach yields performance improvements over vanilla RL, achieving higher pass@1 scores on both MultiPL-E (from 73.6 to 76.3) and the newly introduced CrossPLEval benchmark (from 58.5 to 63.2). The main contributions include a unified on-/off-policy RL framework tailored for multilingual code, a novel group-based preference optimization method leveraging IRs, and the introduction of CrossPLEval, a new benchmark for evaluating cross-lingual code generation.

**Questions:**

- You claim a “significant performance disparity” between popular and low-resource languages motivates your work. How large is this gap after simply continuing SFT on the low-resource slices of public code corpora? Why is a much heavier RL pipeline needed instead of that cheaper remedy?

- Translating between languages does not necessarily expose the model to new algorithmic patterns. What evidence shows translation is causal for closing the cross-language gap, rather than just correlational?

- You frame GEPO as a general technique for structured preference learning. But all experiments are narrowly scoped to IR-based code translation.Have you tested GEPO on any non-code task or natural language reasoning setup?

**Ethical Concerns:**

["NO or VERY MINOR ethics concerns only"]

**Final Justification:**

The authors did not address my concerns sufficiently e.g.
> Question 1: You claim a "significant performance disparity" between popular and low-resource languages motivates your work. How large is this gap after simply continuing SFT on the low-resource slices of public code corpora? Why is a much heavier RL pipeline needed instead of that cheaper remedy?
and
> Weakness 4: If the goal is genuinely closing the gap in resource-poor languages, the more straightforward solution would be to directly generate synthetic data or leverage powerful multilingual pre-trained models (e.g., GPT-4, Code LLaMA), rather than employing complex RL frameworks that require sophisticated translation tasks and fine-grained reward engineering.

Still we don't see a necessity for RL, unless seeing a clear-outperforming results against simple baselines.

**Limitations:**

Yes

**Quality:**

2

**Strengths And Weaknesses:**

## Strength

Under-explored area, efforts to curate the training data and establishing signals would be important to the community.

## Weaknesses
1. The method presupposes a robust suite of cross-language unit tests and IR mappings. In real-world scenarios, such robust testing and IR equivalences are seldom available, especially for less popular languages.

2.  Many programming languages differ dramatically not only in syntax but in execution paradigms, side-effect behaviors, and semantic subtleties (e.g., imperative vs. functional vs. logic-based languages). Representational equivalence at the IR-level might be overly simplistic or even fundamentally incorrect.

3. Missing “realistic” baselines like continuing pre-training on target-language code or retrieval-augmented fine-tuning, which could be cheaper and simpler.

4. If the goal is genuinely closing the gap in resource-poor languages, the more straightforward solution would be to directly generate synthetic data or leverage powerful multilingual pre-trained models (e.g., GPT-4, Code LLaMA), rather than employing complex RL frameworks that require sophisticated translation tasks and fine-grained reward engineering.

---

> ### Author Rebuttal · Authors · 2025-07-29
>
> We sincerely appreciate the **Reviewer ms4a**'s comments and questions. We are glad that **Reviewer ms4a** recognized the importance of our method. Below are responses to address the concerns raised.
>
> ___
> >Weaknesses 1: The method presupposes a robust suite of cross-language unit tests and IR mappings. In real-world scenarios, such robust testing and IR equivalences are seldom available, especially for less popular languages.
>
> We appreciate this concern, and our response clarifies the role of IR data and the generalizability of our method:
> 1.  It's crucial to understand that our method utilizes IR data exclusively during the training stage. During inference, our model operates the same pipelines as existing LLMs on programming benchmarks, **without requiring any IRs**.
> 2.  The core contribution of our method lies in its ability to enhance programming proficiency across diverse languages through code translation tasks and the recognition of IR logical equivalences. This capability is designed to be applicable in real-world scenarios, improving the model's understanding and generation of code in various languages, even those with limited resources.
>
> ___
> >Weakness 2: Many programming languages differ dramatically not only in syntax but in execution paradigms, side-effect behaviors, and semantic subtleties (e.g., imperative vs. functional vs. logic-based languages). Representational equivalence at the IR-level might be overly simplistic or even fundamentally incorrect.
>
> We acknowledge the semantic complexities across different programming paradigms. Our approach to equivalence within GEPO is based on **functional equivalence**. For any two code snippets, if they achieve the same functionality, they are considered equivalent in GEPO. Our algorithm explicitly accounts for subtle differences by minimizing the variance of their corresponding implicit rewards rather than forcing exact equality. Although this doesn't introduce perfect equivalence at a detailed level, our experimental results demonstrate that incorporating this functional-level approximate equivalence significantly improves overall performance.
>
> ___
> >Weakness 3: Missing "realistic" baselines like continuing pre-training on target-language code or retrieval-augmented fine-tuning, which could be cheaper and simpler.
>
> We appreciate the suggestion for additional baselines and would like to clarify our position:
> 1.  Our method **does not utilize LLVM code** during the evaluation stage, making a direct application of RAG less straightforward for comparison. Our primary contribution is the OORL framework, which serves as a general method to enhance the multi-programming language understanding capabilities of Language Models (LLMs). It can be integrated into the training pipeline of any code-domain-specific model.
> 2.  CPT isn't cheaper. Our training process with OORL requires only 4 A100 GPUs for full-parameter fine-tuning over one day, using 2400 samples. This demonstrates a highly efficient training process.
> 3.  SFT on low-resource languages often necessitates extensive data collection, including correct reasoning processes and corresponding code, which incurs substantial human labor. Such datasets typically require large scales (100k+ examples). In contrast, our method leverages unit tests, where test case conversion across languages can be easily **automated via rules**, significantly reducing the data collection burden, especially for low-resource languages.
>
> ___
> >Weakness 4: If the goal is genuinely closing the gap in resource-poor languages, the more straightforward solution would be to directly generate synthetic data or leverage powerful multilingual pre-trained models (e.g., GPT-4, Code LLaMA), rather than employing complex RL frameworks that require sophisticated translation tasks and fine-grained reward engineering.
>
> We argue that our approach offers distinct advantages over direct synthetic data generation:
> 1.  Directly synthesizing or generating high-quality low-resource language data for fine-tuning is resource-intensive, requiring significant human effort and time, and it struggles to cover a wide range of functionalities.
> 2. Our OORL training framework is applied to LLMs after post-training. Consequently, our method does not conflict with the use of the SFT method.
> 2. Moreover, as the multi-programming language understanding improves during RL training, it can itself generate higher-quality low-resource language data during rollouts, which further enhances its training in a self-improving loop. This inherent mechanism makes our method a powerful data augmentation technique.
> 3.  Crucially, our training method demonstrates strong generalization capabilities, improving the model's proficiency even in programming languages not explicitly present in the training set. This is typically challenging for SFT-based approaches, whereas RL algorithms generally exhibit superior generalization.
>
> ___
> >Question 1: You claim a "significant performance disparity" between popular and low-resource languages motivates your work. How large is this gap after simply continuing SFT on the low-resource slices of public code corpora? Why is a much heavier RL pipeline needed instead of that cheaper remedy?
>
> 1.  Even with current code LLMs that have been extensively trained with various programming languages, a substantial performance disparity persists between different programming languages for the same coding requirements (As shown in Table 2 and Qwen2.5 Coder [1]).
> 2.  Our RL pipeline is not "heavier". It is quite efficient. As mentioned previously, we achieved our results with only 4 A100 GPUs and 2400 samples for just one day.
> 3.  Although SFT on low-resource data might reduce the performance gap, it often leads to a degradation in overall performance. Our goal is comprehensive multi-programming language understanding, not merely focusing on low-resource languages. As shown in Tables 2 and 3, our method demonstrates improvements in both widely-used languages and less common ones.
>
> [1] Binyuan Hui, Jian Yang, Zeyu Cui, Jiaxi Yang, Dayiheng Liu, Lei Zhang, Tianyu Liu, Jiajun Zhang, Bowen Yu, Keming Lu, et al. Qwen2. 5-coder technical report. arXiv preprint arXiv:2409.12186, 2024.
>
> ___
> >Question 2: Translating between languages does not necessarily expose the model to new algorithmic patterns. What evidence shows translation is causal for closing the cross-language gap, rather than just correlational?
>
> As detailed in Table 4, we provide empirical evidence that the code translation task is indeed causal in closing the cross-language performance gap. We perform code translation tasks with an RL algorithm and achieve significant improvement. This is one of the key contributions of our work, demonstrating that this specific task design within our framework directly leads to improved multi-programming language understanding.
>
> ___
> >Question 3: You frame GEPO as a general technique for structured preference learning. But all experiments are narrowly scoped to IR-based code translation.Have you tested GEPO on any non-code task or natural language reasoning setup?
>
> 1.  The GEPO algorithm is developed to specifically address the problem of the significant performance disparity between popular and low-resource languages in code. While it is fundamentally a general preference learning algorithm, its application requires defining a specific notion of "equivalence" for the task at hand. In this work, we leverage **code functional equivalence**. Applying GEPO to other natural language tasks would necessitate defining task-specific equivalence concepts, making it a highly specific application rather than something easily testable on general preference learning datasets.
> 2.  Within the context of this paper, we conducted an ablation study comparing GEPO with the DPO algorithm on code-related tasks (Table 4). Our results demonstrate that GEPO outperforms the general DPO algorithm in this domain. From a formal perspective, DPO can be considered a special case of GEPO where the winner group and loser group each contain a single sample.
>
> ___

---

### Official Review · Reviewer_9pZg · 2025-07-03

**Clarity:** 3
**Significance:** 3
**Originality:** 3
**Rating:** 5
**Confidence:** 3

**Summary:**

The paper proposes OORL, a RL framework designed to improve LLMs' understanding of code across diverse programming languages, especially in low-resource scenarios. OORL combines on-policy RL with rule-based binary rewards from unit tests and a new off-policy method called Group Equivalent Preference Optimization (GEPO). GEPO guides LLMs to distinguish functionally equivalent IRs from non-equivalent ones by comparing groups of IRs, leveraging fine-grained equivalence signals. This enables models to better generalize code translation and generation across languages.

**Questions:**

What does the learning curve of the proposed RL training look like? Does the good performance emerge gradually or appear suddenly?

**Ethical Concerns:**

["NO or VERY MINOR ethics concerns only"]

**Final Justification:**

I don’t see significant weaknesses in this paper, and the authors have addressed my question regarding the experiments.

**Limitations:**

I don't see obvious negative societal impact of this work.

**Paper Formatting Concerns:**

No Paper Formatting Concerns

**Quality:**

3

**Strengths And Weaknesses:**

**Strengths**
* The research problem of code translation is practically useful. For example, it can accelerate system upgrades of old industrial codebases written in legacy or low-resource languages, such as Bash, Haskell, by translating them into modern languages like Python or C#.
* The IR-based reward design is well-motivated and reasonable. It provides a good way to incorporate functional alignment into RL training beyond surface forms, and the design is intuitive and grounded in the nature of programming languages.
* The experimental results support the claims, demonstrating significant improvements in code translation between four popular languages and a set of target languages, including low-resource ones.


**Weaknesses**

I don’t see significant weaknesses.

---

> ### Author Rebuttal · Authors · 2025-07-29
>
> We sincerely appreciate the **Reviewer 9pZg**'s insightful comments and questions. We are glad that **Reviewer 9pZg** recognized our useful and well-motivated method and comprehensive experiments. Below are responses to address the question raised.
>
> ___
> >Questions 1: What does the learning curve of the proposed RL training look like? Does the good performance emerge gradually or appear suddenly?
>
> The learning curve of our RL training exhibits the following behavior:
> 1. Performance shows a sharp increase at the beginning. This initial surge is primarily driven by the model quickly learning the desired response format and basic task structure.
> 2. Following the initial rapid increase, the performance continues to improve steadily but more slowly.
> 3. Finally, the learning curve gradually stabilizes and converges within a certain range, indicating that the model has reached a plateau in its learning for the given task.
>
> ___

---

> > ### Comment · Reviewer_9pZg · 2025-08-08
> >
> > Thank you to the authors for your response! I’m keeping my positive score of 5.

---

### Official Review · Reviewer_xoTJ · 2025-07-03

**Clarity:** 4
**Significance:** 3
**Originality:** 3
**Rating:** 5
**Confidence:** 4

**Summary:**

In the present paper, the authors present OORL a combination of on-policy RL in the form of Reinforce++, and off-policy RL with group equivalent preference optimization. Applied to code, the authors leverage the underlying compiler infrastructure in the form of language-independent compiler representations to extend the code model's performance to lower-resource languages such as C#. The main novelty of the paper lies in the development of the GEPO off-policy approach, and how it leverages compiler IR. The performance of the approach is proven across a number code translation benchmarks on which the proposed approach outperforms other contemporary models.

**Questions:**

- Compiler intermediate representation themselves have a heavy distribution bias towards certain programming languages such as C++, both with regards to the length and amount of IR available per language. How do you seek to remedy, or potentially even exploit, this in practice?

- The lines of code in the average code sample contained in the evaluation are fairly short, and may not be as representative as real world code. Do the authors imagine it to be possible to construct a "hero"-showcase where longer forms of code (& IR) are used to better illustrate the approaches' limitations in practice. Particularly this quote from the limitations "GEPO manages larger groups of preference data, necessitating more GPU memory resources. This increased memory can be particularly pronounced with larger group sizes or longer response sequences." gives me pause of concern here.

- Have the authors' considered how their approach could potentially also scale to more HPC-focussed languages? For example, there exist the Joseph Burkardt database of Fortran routines <https://people.sc.fsu.edu/~jburkardt/f77_src/f77_src.html> for which there are also direct function implementations in other languages such as C, C++, Java, and more available. Evaluating on a subset of the database would provide ample evaluation of the approaches' performance on HPC-centric programming languages.

- Do the authors know whether their approach is sensitive to different generations of LLVM intermediate representation? I.e. LLVM 16 vs 17 vs 18 vs 19 vs 20. Have the authors for example tested their approach with different version of the LLVM compiler to make sure that the evaluation performance stays constant? Testing even with one different LLVM version would aid in assessing the approaches' sensitivity to this compiler specification.

- As is, the potential of larger models, when prompted correctly, is not considered here at all. Have the authors considered to for example prompt larger models such as Claude Opus, or DeepSeek R1 to perform the same task? Adding this point of comparison would aid greatly in assessing the paper's contributions more comprehensively.

**Ethical Concerns:**

["NO or VERY MINOR ethics concerns only"]

**Final Justification:**

I firmly recommend the acceptance of the present work as I believe for it to be a significant advancement towards extending machine learning for code to lower resource languages, a pressing issue of modern models.

Specifically, the following points made me arrive at my final score:
- Algorithmic clarity, in the review especially, the authors managed to provide further clarity around training- vs inference-time usage of componentry. This addressed questions around the usage of IR, as well as around the memory footprint of the proposed approach.
- Further discussion on the distinction between the presented approach, and the scale its evaluations is connected, and the inherent motivation behind it.

As such I wholeheartedly recommend the acceptance of this work.

**Limitations:**

yes

**Paper Formatting Concerns:**

The paper has a number of typos in its writeup, such as e.g.
- Figure 2, "equivalenct" -> "equivalent"

Going over the paper again to weed out these last spelling errors, would aid the paper greatly.

**Quality:**

3

**Strengths And Weaknesses:**

The paper has its main strengths in its intellectual clarity, its clear derivation of its optimization objectives enabling reproducibility with a reasonable amount of effort, and its style of evaluation. Especially its evaluations cover a wide range of programming languages, and as such provide a fair assessment of the approaches' strength across programming languages.

The main weaknesses of the paper break down into the following points:
- The average lines of code per function, i.e. table 1, is too short for a representative benchmark. Especially the length does seem to not be representative of production code. See e.g. ComPile (https://huggingface.co/datasets/llvm-ml/ComPile), figure 3 for a more typical distribution of the length of production code when looking at the total instruction count.
- The authors point out in the limitations, that GEPO occupies a lot of memory due to its group equivariant nature. Continuing on from the previous point, the paper fails to show when this limitation would apply in practice. Even considering a synthetic example to illustrate the limitations in terms of the length of code would improve the paper greatly here.
- The evaluation only considers a highly limited number of models. Large language models which have seen LLVM IR during training such as StarCoder 2, or Meta's LLM-Compiler are not included in the evaluation, and modern large models such as Claude Opus are not considered in their entirety. As stands, the evaluation fails to capture these points of comparison and makes it hard to place its performance in relation to these models.

---

> ### Author Rebuttal · Authors · 2025-07-29
>
> We sincerely appreciate the **Reviewer xoTJ**'s insightful comments and questions. We are glad that **Reviewer xoTJ** recognized the innovation of our method and comprehensive experiments. Below are responses to address the concerns raised.
>
> ___
> >Weaknesses 1 & Questions 2:
>
> We appreciate your comments regarding the representativeness and length of our benchmark. We'd like to offer the following clarifications:
> 1.  **Benchmark Representativeness**: As shown in Table 1, our newly proposed benchmark **CrossPLEval**, features an average of 34.57 Lines of Code (LoC) per function. This is approximately 5 times longer than traditional MultiPL-E benchmarks, marking a significant step towards more realistic code samples while remaining within a scope that allows for meaningful performance evaluation with current LLMs. Datasets that are excessively complex can sometimes obscure subtle but important improvements, especially given the current capabilities of models.
> 2.  **Focus on Function-Level Evaluation**: Our evaluation primarily uses code samples at the function level. This was a deliberate choice to align with our core objective: to verify whether our proposed GEPO method can effectively enhance a model's deep understanding of code's "functional semantics" by learning the equivalence and non-equivalence of IRs. Functions are the smallest, self-contained units that encapsulate specific functionalities. Evaluating at this level provides the most direct and clear means to measure our method's ability.
> 3.  **ComPile Dataset**: We acknowledge ComPile as a valuable IR translation dataset. It could potentially be used for training GEPO by constructing positive and negative groups, but it is not ideal for our evaluation, as our tasks involve high-level code generation. **Because our model operates the same pipelines as existing LLMs on programming benchmarks without requiring any IRs during inference**.
> 4.  **GEPO Memory Usage Clarification**: It's crucial to clarify that GEPO's additional GPU memory resources are primarily required during training, not during inference. The increased memory consumption in training stems from managing larger groups of preference data. Although this is an overhead, it leads to increased data utilization efficiency. Importantly, the trained model, once deployed, operates with standard memory footprints, making it viable for practical application on longer code samples without additional memory concerns.
>
> ___
> >Weaknesses 2:
>
> 1. As previously mentioned, GEPO's additional memory occupation occurs exclusively during the training stage.
> 2.  The increased computational overhead during training is a direct consequence of the enhanced data utilization provided by GEPO. By processing groups of equivalent and non-equivalent IRs, GEPO extracts more robust and nuanced semantic understanding. From a data efficiency perspective, this trade-off is well-justified by the significant improvements in model performance and its ability to generalize across programming languages.
>
> ___
> >Weaknesses 3:
>
> We appreciate your desire for a broader comparison against more diverse models. Our evaluation strategy is guided by specific objectives:
> 1.  Our proposed OORL framework is validated on models within the 7B parameter scale. Direct comparisons with significantly larger models, such as Claude Opus, which operate at a much larger scale, are not directly comparable given the vast difference in model sizes and computational resources required. Our focus is on demonstrating the effectiveness of GEPO as an optimization method for LLMs, rather than on achieving new SOTA results with the largest models.
> 2.  Our primary goal is to enhance multi-programming language understanding capabilities. **We do not utilize LLVM as a reference input for the model during the evaluation process**. Consequently, models like StarCoder2 and LLM-Compiler do not consistently demonstrate competitive performance when evaluated on our specific high-level code generation tasks. We strategically chose the strongest available open-source models of comparable size to rigorously demonstrate the effectiveness of our proposed method in improving code understanding and generation across diverse languages.
>
> ___
> >Questions 1:
>
> We are grateful for this insightful question regarding the inherent distribution bias of compiler IRs. **Reviewer xoTJ** accurately highlights that the current compiler IR ecosystem is heavily structured around languages like C/C++. We wish to clarify that this focus is **not an oversight, but a conscious methodological choice, which we indeed seek to exploit.**
>
> Our core idea is to leverage the high-quality IR generated from resource-rich languages such as C/C++ as an ideal **knowledge bridge**. As we discuss in the paper, compiler IR possesses language-agnostic properties, making it an excellent medium for aligning the semantics of diverse programming languages. By training our model to learn the mapping from C code to its various functionally equivalent IR forms, we enable the model to grasp a deeper, function-level logic that is independent of any specific language syntax.
>
> Once this abstract understanding of "functional equivalence" is established, it can effectively **generalize to other programming languages**, particularly those low-resource languages that often lack mature IR toolchains. As shown in Table 2 and 3, our method significantly improves code understanding and generation capabilities across a variety of languages, including low-resource languages Bash and Haskell, which are not explicitly present in the GEPO training phase.
>
> ___
> >Question 3:
>
> The core mechanism of our framework relies on learning functional semantics captured from compiler Intermediate Representations (IRs), rather than being tied to the syntax of specific source languages. While our current evaluation focuses on a range of mainstream and lower-resource languages, there are **no fundamental barriers that would prevent its application to HPC languages**, such as Fortran.
>
> Our method has already demonstrated strong generalization capabilities to programming languages that were not explicitly included in the training set (e.g., Bash and Haskell). This suggests that the framework's learned understanding of functional equivalence can effectively extend to other languages, including those relevant to HPC, without requiring direct training on them. This scalability to new domains is a key strength of our IR-based approach.
>
> Indeed, the Joseph Burkardt database is an excellent resource, and evaluating on a subset of it would be a valuable avenue for future work.
>
> ___
> >Questions 4:
>
> We appreciate this insightful question regarding the sensitivity of our approach to different LLVM IR versions.
> 1.  It is crucial to clarify that our **test process does not involve the use of LLVM** for inference. The model, once trained, directly generates high-level code based on a high-level code prompt. Consequently, the evaluation performance is **not affected by the specific LLVM version** during testing, **as LLVM is not part of the inference pipeline**.
> 2.  The enhancement in our model's capabilities primarily stems from its learning of functional semantics directly from the compiler's IRs **during the training phase**, coupled with the multi-programming language translation tasks. While different LLVM versions might produce slightly varied IRs for the same source code, our method's strength lies in learning the underlying *equivalence relations* between these IR forms. This indicates that the core knowledge acquisition happens at the training stage, making the inference robust to variations in LLVM versions during evaluation. For future work, we could explore training on data generated from multiple LLVM versions to further enhance robustness, but for the current study, the inference process remains unaffected.
>
> ___
> >Questions 5:
>
> We acknowledge the value of comparing with larger, state-of-the-art models for a comprehensive assessment.
>
> Our paper primarily introduces the OORL framework as a post-training method for LLM training, aimed at enhancing multi-programming language understanding capabilities. The framework can be applied to any code-domain-specific model to improve its code understanding. Naturally, larger models (Claude4 Opus, DeepSeek R1, etc.) with significantly more parameters and extensive pre-training are expected to outperform our 7B-scale model in absolute terms.
>
> Regarding prompting, as clarified previously, our testing phase does not involve the use of LLVM code. Therefore, there are no specific "special prompt tricks" related to LLVM. We utilize a standard evaluation pipeline [1, 2], ensuring a fair comparison.
>
> [1] An Yang, Anfeng Li, Baosong Yang, Beichen Zhang, Binyuan Hui, Bo Zheng, Bowen Yu, Chang Gao, Chengen Huang, Chenxu Lv, et al. Qwen3 Technical Report. arXiv preprint arXiv:2505.09388, 2025.
>
> [2] Aixin Liu, Bei Feng, Bing Xue, Bingxuan Wang, Bochao Wu, Chengda Lu, Chenggang Zhao, Chengqi Deng, Chenyu Zhang, Chong Ruan, et al. DeepSeek-V3 Technical Report. arXiv preprint arXiv:2412.19437, 2024.
>
> ___
> >Paper Formatting Concerns 1:
>
> Thanks for meticulously pointing out the typographical errors. We sincerely apologize for these oversights. We will conduct a thorough review of the entire paper to correct all spelling and grammatical errors, ensuring a polished and professional final version. Your attention to detail is greatly appreciated.
>
> ___

---

> > ### Comment · Reviewer_xoTJ · 2025-08-05
> > **Thank You**
> >
> > I thank the authors for their thoughtful, and detailed response. I especially appreciate the further clarification around the demands posed by the method on training time vs at inference time. As such I have raised my score.

---

> > > ### Author Response · Authors · 2025-08-05
> > >
> > > Thanks for the positive feedback. We are delighted that our response addressed the concerns.

---

### Decision · Program_Chairs · 2025-09-17

**Decision:**

Accept (poster)

**Comment:**

This paper uses the code translation task (between different programming languages) to improve code generation. The reviewers seem to think that the paper has a significant value, and that the claims are strong. The authors have answer questions meticulously.